# SDRC-YOLO: A Novel Foreign Object Intrusion Detection Algorithm in Railway Scenarios

Caixia Meng [1,2], Zhaonan Wang [3,*], Lei Shi [1,3,4], Yufei Gao [3,4], Yongcai Tao [1,4] and Lin Wei [3,*]

[1] School of Computer and Artificial Intelligence, Zhengzhou University, Zhengzhou 450001, China
[2] Department of Image and Network Investigation Technology, Railway Police College, Zhengzhou 450053, China
[3] School of Cyber Science and Engineering, Zhengzhou University, Zhengzhou 450002, China
[4] Songshan Lab, Zhengzhou 450046, China
[*] Correspondence: wangzhaonan@gs.zzu.edu.cn (Z.W.); weilin@zzu.edu.cn (L.W.)

**Abstract:** Foreign object intrusion detection is vital to ensure the safety of railway transportation. Recently, object detection algorithms based on deep learning have been applied in a wide range of fields. However, in complex and volatile railway environments, high false detection, missed detection, and poor timeliness still exist in traditional object detection methods. To address these problems, an efficient railway foreign object intrusion detection approach SDRC-YOLO is proposed. First, a hybrid attention mechanism that fuses local representation ability is proposed to improve the identification accuracy of small targets. Second, DW-Decoupled Head is proposed to construct a mixed feature channel to improve localization and classification ability. Third, a large convolution kernel is applied to build a larger receptive field and improve the feature extraction capability of the network. In addition, the lightweight universal upsampling operator CARAFE is employed to sample the size and proportion of the intruding foreign body features in order to accelerate the convergence speed of the network. Experimental results show that, compared with the baseline YOLOv5s algorithm, SDRC-YOLO improved the mean average precision (mAP) by 2.8% and 1.8% on datasets RS and Pascal VOC 2012, respectively.

**Keywords:** railway intrusion detection; hybrid attention; Decoupled Head; super-large convolution kernel; upsampling operator





## 1. Introduction

Foreign objects intruding into the railway system pose a great threat to railway operations. The detection and identification of intrusions have been the focus of significant research. An effective intrusion detection method is of great significance to railway security.

At present, there are two main methods of railway intrusion detection: contact detection and non-contact detection [1]. Contact detection, which is not suitable for large-scale use, requires significant engineering work and is troublesome to install [2]. At the same time, if no manual disposal is carried out in time when intrusions are detected, the running efficiency of the train will be affected [3,4]. Deep learning-based detection is an effective method for non-physical contact intrusion detection, and there is a lot of related work in the railroad scenario [5]. It has the advantages of easy maintenance and intuitive results [6]. However, the complexity and variability of the railroad environment and the interference of bad weather can lead to misreporting. High detection accuracy and a low false positive rate may be, achieved to some extent, through the development of deep learning algorithms. Hussain, R. et al. [7] use the reverse attention mechanism for salient object detection to accomplish clear boundary detection of objects; Wang et al. [8] propose a new unsupervised intrusion detection method based on deep generative networks and autoregressive models, which can effectively detect anomalies. Deep learning methods, however, are slow and use a large amount of memory, calling for high-performance computers. Therefore, for complex

railroad scenarios, an efficient railroad intrusion detection method with more balanced accuracy and speed is needed.

SDRC-YOLO is proposed as a solution to these issues. SDRC-YOLO is an enhanced version of the YOLOv5s algorithm which can successfully identify infiltration in challenging railway circumstances. The following are the primary contributions of this paper:

(1)  An SSA hybrid attention mechanism with a spatial attention module (SAM) working in tandem with Squeeze and Excitation Network (SENet) channel attention is proposed. This mechanism effectively improves local representation ability, integrates multiple receptive fields, enriches the information, makes the characteristics cover more of the intrusion, better fits the relevant characteristics information, enhances the attention to small targets, and improves the effectiveness of the model in small target recognition;

(2)  An efficient decoupled head based on a hybrid channel strategy is proposed and called the DW-Decoupled Head. Compared with the non-decoupled end-to-end method, the DW-Decoupled Head can speed up the network convergence, reduce the computational cost, and achieve faster inference speed;

(3)  Large convolutional kernels RepLKNet with heavy parameterization are adopted to build larger perceptual fields. In addition, the lightweight universal upsampling operator CARAFE is used for sampling, and more suitable sizes and ratios for the characteristics of intrusion are adopted;

(4)  An RS railway intrusion dataset is built. There is no public railway intrusion dataset at present. As pedestrians, cars, and bicycles are the most frequent intrusions, they were chosen as the main study subjects.

## 2. Related Work

Railway security is an ongoing study area for intrusion detection. Contact detection method mainly uses sensing technology to judge whether the foreign object is in contact with the sensor. This physical protection method is greatly affected by the outside factors, with high costs. It cannot provide early warning for an intrusion in which the foreign object does not come into contact with the sensors. On the other hand, convolutional neural network-based target detection methods, including one-stage and two-stage algorithms, are popular non-contact intrusion detection techniques in railway applications. One-stage algorithms are fast in detection process from the input image to the output prediction results. However, while the anchor mechanism generates many candidate boxes, few of them are beneficial for detection: candidate boxes reduce the performance of the detection algorithm. Common one-stage algorithms include YOLO [9–11], SSD [12], RetinaNet [13], EfficientDet [14], etc. For input, two-stage methods create a number of candidate boxes, which are then classified and regressed. The detection speed is slow since it requires multiple detection, classification, and correction processes, but the network performance is high. Common two-stage algorithms include R-CNN [15], Fast R-CNN [16], Faster R-CNN [17], etc. The conventional target identification technique, which is based on a convolutional neural network, has many parameters and operates slowly in real-world scenarios.

Many scholars have studied railway intrusion detection based on deep learning methods. He and Ren et al. [18] proposed a train obstacle detection method based on improved R-CNN. A new parallel upsampling structure and context extraction module were added to the architecture, increasing the accuracy of R-CNN to 90.6%. To strengthen the capability to recognize small target objects, He et al. [19] applied the enhanced Mask R-CNN rail transit obstacle detection approach, suggested a new feature extraction network, and incorporated a number of multi-scale improvement techniques. The two-stage target detection technique is the foundation of all the approaches mentioned above. Target identification can be enhanced to some extent by extending the sensitivity field and merging shallow and deep features for multi-scale feature fusion. Real-time detection, however, is not possible because of the region proposal network. Zhang et al. [20] proposed a detection algorithm for high-speed rail intruders based on the YOLOv3 network. By improving the

FPN structure and using switchable hollow convolution to extract features, the algorithm reduced false positives in target detection, but the frame per second (FPS) was low and could not meet the requirements of real-time detection. Ref. [21] proposed a lightweight adaptive multi-scale feature fusion object detection network based on YOLOv3 to improve the performance of small target detection in complex environments. This work is based on the YOLOv3 network architecture, while the YOLOv5 algorithm has better performance and faster speed and is widely used in the industry. Thus, research can be continued based on the YOLOv5 network structure to obtain the most suitable railway intrusion detection algorithm for complex environments and improve detection performance.

In this paper, a better railway intrusion detection approach based on the YOLOv5 network is presented in order to fulfill the real-time and accuracy requirements of railway intrusion detection. As one of the mainstream single-phase networks, YOLOv5 performs well in both detection accuracy and detection speed. The input end, the backbone network, the neck feature fusion, and the YOLO detection head make up the bulk of the YOLOv5 network design. Mosaic data are used to improve the input end of YOLOv5. Four randomly selected images were randomly cropped and then spliced into images with the specified resolution. The backbone network uses the CSPDarknet53 [22] architectural network for feature extraction, which dramatically reduces parameters and computation and boosts performance. A feature pyramid network and a path aggregation network (PANet) [23] are used in the neck section. To create a feature pyramid structure with varied sizes that collects rich feature information, in-depth features taken from the trunk network are upsampled and added to the shallow information element by element. The three detection layers of the YOLO detection head are responsible for predicting large targets, medium targets, and small targets, respectively. The results of detection include target location information, object information, and category information. In this paper, the output feature mode, receptive field range, and upsampling mode of the YOLOv5 network architecture are improved. These changes can help minimize the misreporting rate and missing report rate of railway intrusion detection while enhancing the algorithm's capacity to extract characteristics for distant and small intrusions.

## 3. SDRC-YOLO Network Model

Based on the YOLOv5s network, an SSA hybrid attention mechanism in which channel attention is connected with spatial attention is used in the SDRC-YOLO algorithm. Global contextual information can be extracted to enhance the mining of small target features and improve the recognition of small targets in images. A more efficient DW-Decoupled Head detector is used to extract useful location information, and a hybrid channel strategy is used to build a more efficient decoupled head to speed up network convergence. RepLKNet with a super-large convolutional kernel is used to widen the receptive field. The lightweight upsampling operator CARAFE is used to optimize the upsampling method, as its sampling is more suitable for the size and ratio of the characteristics in railway intrusions. The SDRC-YOLO network structure is shown in Figure 1. 'S' stands for SSA Hybrid Attention Mechanism, 'D' stands for DW-Decoupled Head, 'R' stands for Large Convolutional Kernel RepLKNet, and 'C' stands for the upsampling operator CARAFE.

### 3.1. SSA Hybrid Attention Mechanism

The YOLOv5 feature fusion route aggregate network's convolution layer has hundreds of channels, rendering the important characteristics invisible. An SSA hybrid attention module is suggested in order to strike a compromise between weight and model accuracy. Weight information is added to the channels and geographical locations of the feature map through an understanding of the significance of various channels, alongside spatial position information and target category information. In this way, the features can cover more parts of the object to be recognized. The degree of attention to small targets is enhanced, and the recognition effect on the small target is improved. The two components of the hybrid attention module are the spatial attention module (SAM) [24] and the Squeeze-

and Excitation Network (SENet) [25]. The attention mechanism is implemented in the spatial dimension and channel dimension of the feature map, respectively. The SSA hybrid attention module used in this article is shown in Figure 2.

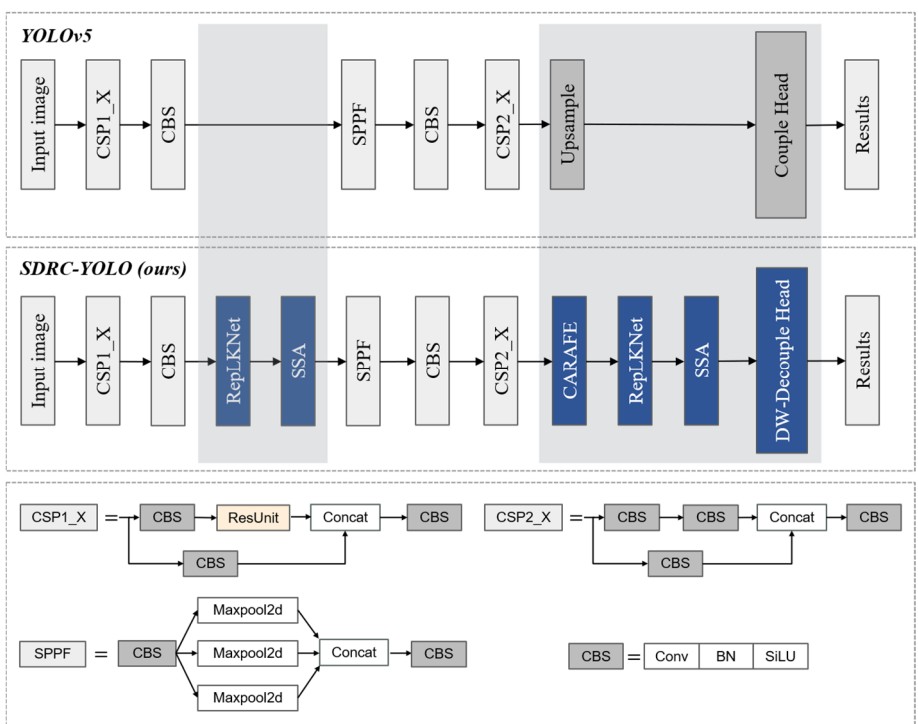

**Figure 1.** SDRC-YOLO network structure.

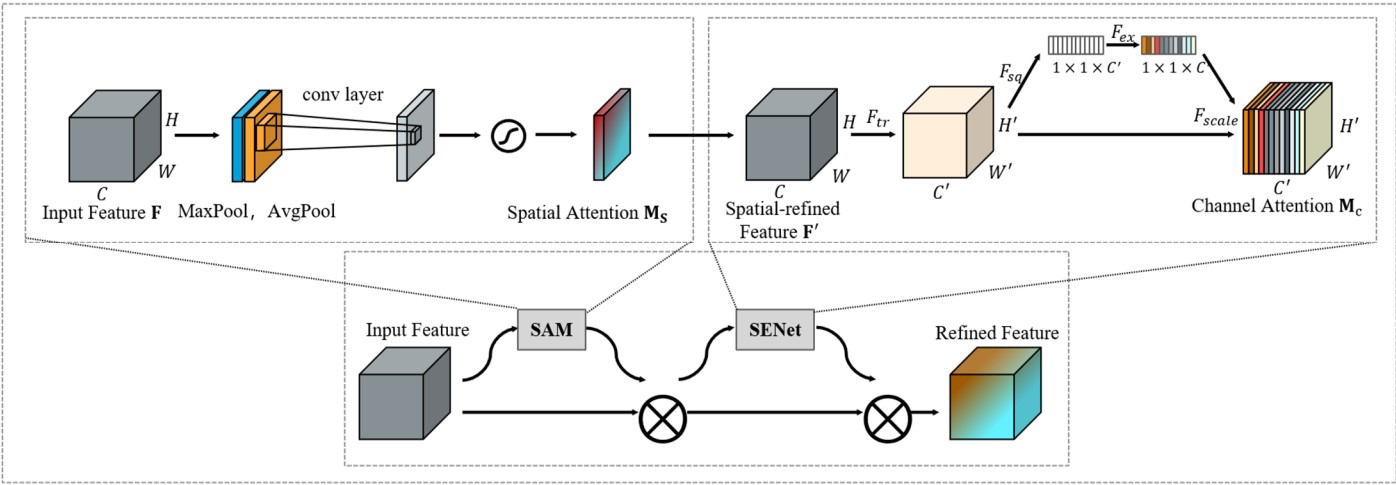

**Figure 2.** SSA hybrid attention module.

In Figure 2, $F_{tr}$ denotes the convolution operation, $F_{ex}$ denotes the excitation mapping, $F_{sq}$ denotes the compression mapping, and $F_{scale}$ denotes the channel-by-channel weighting given to the previous features by multiplication.

SAM combines the channel-wide average pooling and maximum pooling procedures into a single feature map channel, which can be formulated as

$$
\begin{aligned}
M_s(F) &= \sigma\left(f^{7\times7}([\text{AvgPool}(F); \text{MaxPool}(F)])\right)\\
&= \sigma\left(f^{7\times7}\left(\left[F_{avg}^s; F_{max}^s\right]\right)\right)
\end{aligned}
\tag{1}
$$

where $\sigma$ denotes the sigmoid activation function and $f^{7\times7}$ denotes the convolution kernel with a size of $7 \times 7$.

Squeeze and Excitation are the two phases that make up SENet, which is a supplement to the spatial attention module. To obtain the global compression feature vector of the current feature map, the Squeeze phase performs the average pooling operation on the feature map layer. The weight for each channel in the feature map is obtained by using the whole connection on both levels. The weighted feature map serves as the network's next layer's input. The Squeeze operation is mathematically defined in Equation (2). The Excitation operation is shown in Equation (3):

$$F_{sq} = \frac{1}{H \times W} \sum_{i=1}^{H} \sum_{j=1}^{W} u_c(i, j) \tag{2}$$

where $u_c$ represents the set of local descriptors and the statistical data of descriptors can express the whole image.

$$F_{ex}(z, W) = \sigma(g(z, W)) = \sigma(W_2 \delta(W_1 z)) \tag{3}$$

Here, $\delta$ denotes the ReLU activation function, $W_1$ and $W_2$ are weight matrices of the two fully connected layers, and the statistic z is generated by the output feature graph of the spatial dimension $H \times W$ contraction. The output results of the convolutional layer are first weighted by the spatial attention module and then weighted by a spatial attention module. SSA hybrid attention allows the global information to play a greater role, better fits the relevant feature information, and improves the recognition effect on small targets.

### 3.2. DW-Decoupled Head

Most of the existing target detectors use coupled detection heads for classification and regression prediction of the same output features, but the conflict between classification and regression tasks is a well-known problem. The task of target identification is more challenging under the interference of complex railway backgrounds and multi-class targets. Occluding objects requires accurate positioning information to determine their respective positions. In order to improve the accuracy of location and classification, it is proposed that a DW-Decoupled Head be used to predict the target and decouple a separate feature channel from the location and classification task for the boundary box coordinate regression and object classification.

As shown in Figure 3, in order to limit the number of parameters, the DW-Decoupled Head performs $1 \times 1$ convolution on the feature map's channel dimension of the input feature map. The feature map's output is split into two branches. One branch is the classification of a task. It extracts features by $3 \times 3$ depth-wise convolution and adjusts the channel dimensions of the feature map to the number of categories of the predicted target by $1 \times 1$ convolution. The other branch is the positioning of the task. It extracts features by $3 \times 3$ depth-wise convolution. Then, the feature map is divided into two parts: one predicts the central coordinates of the boundary box and the height and width of the box $\{x, y, w, h\}$; the other obtains the confidence score of the target to judge the intersection ratio between the real target box and the predicted box.

Compared with the coupled detection head, which directly integrates multiple pieces of information into one feature map, the DW-Decoupled Head can effectively avoid conflicts between different feature information required by different tasks, and enhance positioning and classification. At the same time, the decoupling head can retain the information of each channel through the operation of depth and breadth and reduce the calculation cost.

### 3.3. Super-Large Convolutional Kernel Model: RepLKNet

In the past work, large convolution kernels used in CNN networks had little currency, and most works used many small convolution kernels for stacking to expand the receptive field. Large core convolution is good at extracting large receptive field features, while small

core convolution is good at extracting small receptive field features. The features of the large receptive field and small receptive field are complementary, and feature extraction can be effectively improved by mixing the features of different receptive fields. The CNN network can learn relevant texture information and shape information, but for humans, the identification of different targets relies more on shape information. Therefore, if a model can capture stronger shape bias information, it will be more friendly to target detection tasks; furthermore, improving the convolution core size can well improve CNN's ability to capture shape bias. To address small convolution kernels' weakness in extracting shape bias information, this paper introduces a model RepLKNet [26]. Using a large convolutional kernel, a small number of large convolution cores are used to replace the stacking of small convolution cores and a heavily reparameterized large convolution-assisted residual structure is used to establish a large receptive field and improve the network feature extraction ability. The RepLKNet architecture diagram is shown in Figure 4.

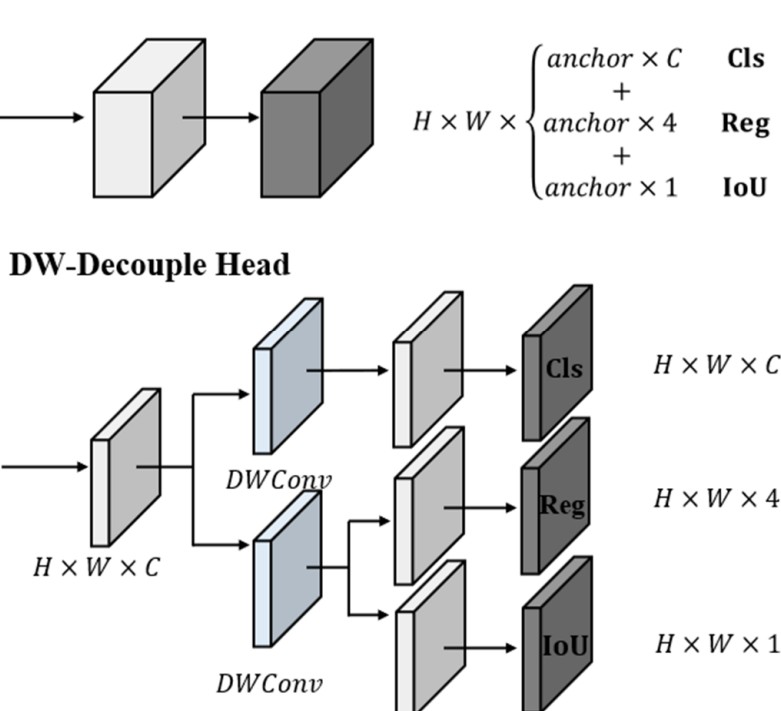

**Figure 3.** Schematic diagram of the coupled head and the DW-Decoupled Head.

In the stem module, the input features are first subsampled twice by the conventional convolutional Conv, and then the lower information is captured by $3 \times 3$ depth-wise Conv (DWConv). Each DWConv uses $5 \times 5$ cores for reparameterization, and $1 \times 1$ Conv is used to increase the channels to achieve information exchange between channels.

Each stage consists of multiple RepLK blocks and ConvFFN. RepLK blocks use the shortcut residue structure to densely connect the deepened network layer and limit the number of output channels in each layer to reduce the parameters and computational complexity. The ConvFFN part is composed of $1 \times 1$ Conv, a GELU activation function, and a BN layer. Compared with the LN (layer normalization) layer used by FFN before the FC layer, the BN used by ConvFFN can be added to the Conv to improve the reasoning speed.

The transition module is placed between different stages to adjust the feature resolution and the number of channels. It is composed of $1 \times 1$ Conv and $3 \times 3$ DWConv. First, $1 \times 1$ Conv is used to improve the channel dimension; then DWConv with a step size of 2 is used for downsampling.

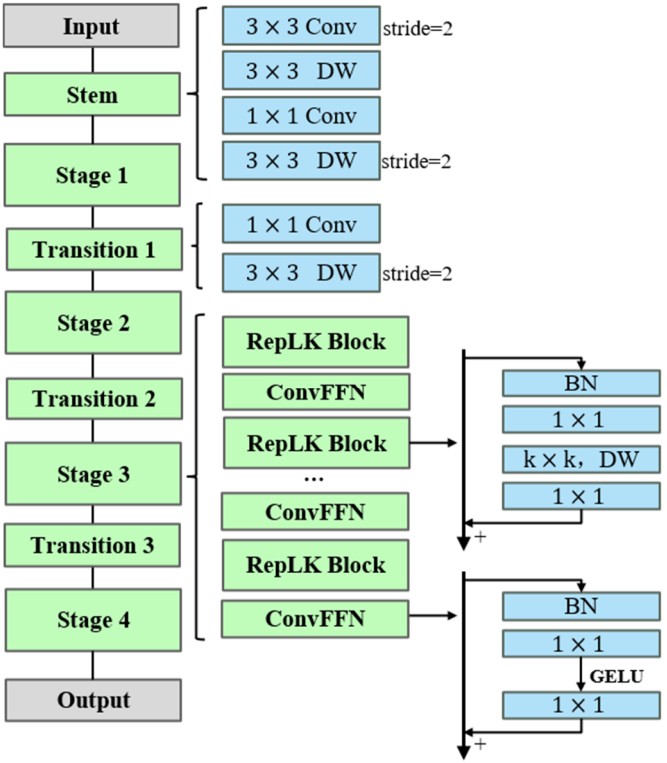

**Figure 4.** Schematic diagram of the RepLKNet architecture.

*3.4. Lightweight Upsampling Operator CARAFE*

YOLOv5 adopts the nearest neighbor interpolation method for upsampling, which applies the same upsampling kernel to every position of the feature graph. The feature graph's semantic information is not used in the upsampling process, despite the approach's speed and the limited number of parameters, which results in substantial feature loss. The lightweight upsampling operator CARAFE [27] is used to address the aforementioned issues and the structure diagram is shown in Figure 5.

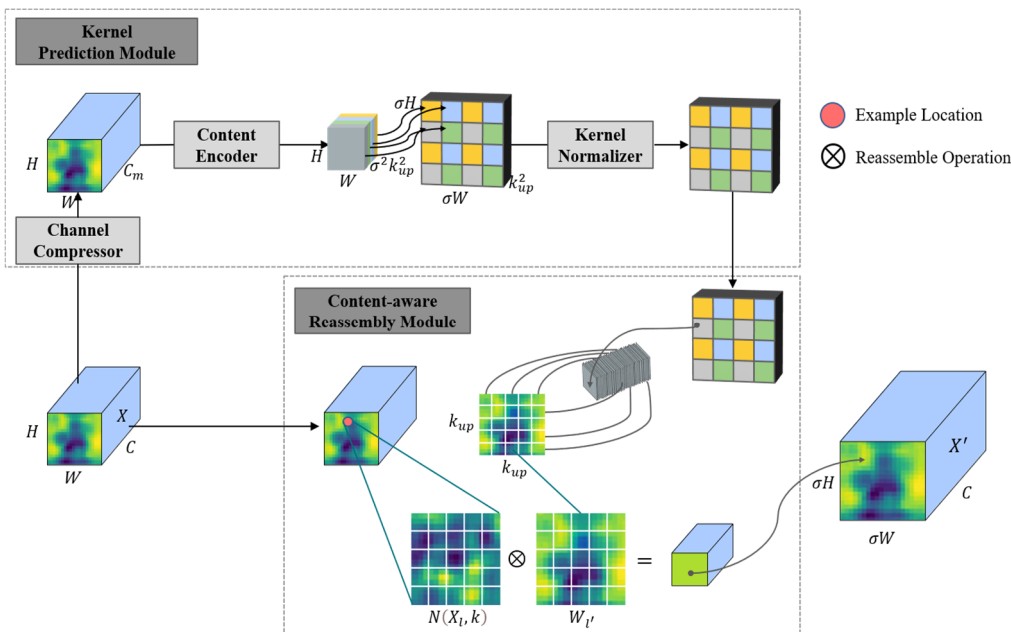

**Figure 5.** Schematic representation of the CARAFE upsampling operator structure.

CARAFE consists of two main modules: an upsampled kernel prediction module and a feature recombination module. Assume that the upsampling multiplier is σ and the upsampling kernel size is $k_{up} \times k_{up}$, and an input feature graph of $H \times W \times C$ is given. In the upsampling kernel prediction module, the input feature graph's channel count is compressed to $H \times W \times C_m$ through a $1 \times 1$ convolution, and the convolution layer of $k_{encoder} \times k_{encoder}$ is then used to forecast the upsampled kernel for the reduced input feature graph. The upsampled kernel with the shape of $\sigma H \times \sigma W \times k_{up}^2$ is obtained by expanding the channel dimension in the spatial dimension from the $C_m$ number of input channels to the $\sigma^2 k_{up}^2$ number of output channels. Feature recombination module: the upsampled kernel obtained in the previous step is normalized so that the weighted sum of the convolution kernel is 1. The input feature map is transferred back to each position in the output feature map, and the region centering on $k_{up} \times k_{up}$ is removed; then, the output feature map of shape $\sigma H \times \sigma W \times C$ is obtained by dot product with the predicted upsampled kernel of the point.

$$W_{l'} = \psi(N(X_l, k_{encoder}\,)) \tag{4}$$

$$X'_{l'} = \phi\big(N(X_l, k_{up}), W_{l'}\big) \tag{5}$$

For any target position $l' = (i', j')$ on the output feature graph $X'$, there is a corresponding source position $l = (i, j)$ on the input feature graph X. Here, $N(X_l, k)$ is represented as the subregion $k \times k$ centered on position l, that is, the adjacent position of $X_l$. As shown in Equation (4), the kernel prediction module ψ predicts the position kernel $W_{l'}$ of each position $l'$ according to the adjacent positions of $X_l$. The feature reorganization module is shown in Equation (5), where φ is the content awareness recombination module, which recombines the features of the adjacent positions of $X_l$ and the core $W_{l'}$.

Compared with the nearest neighbor interpolation method, the following benefits pertain to the lightweight upsampling operator CARAFE: the upsampling core is related to the semantic information of the feature map and is upsampled based on the input content; it is lightweight, without adding additional parameters and calculation; and it has a bigger receptive field and can utilize the surrounding information better. In conclusion, the feature map obtained from CARAFE upsampling is more delicate, losing less detail. The size and proportions of the sampled anchor box are more suitable for the characteristics of railway track intrusion detection.

## 4. Experimental Results

### 4.1. Experimental Setting

The experimental environment used in this work was Python 3.8, Pytorch 1.11.0, CUDA 11.2, and NVIDIA A10-24G. Hyperparameters were hyp.scratch-low. The batch size was set to 32, with a total of 200 training rounds. At the same time, in order to accelerate the network convergence, the official pretraining weight was used as the initial weight, and the initial learning rate of the network training was set to 0.01. The momentum factor was set to 0.937 and the weight attenuation was set to 0.0005.

### 4.2. Experimental Dataset

We experimentally validate the effectiveness of the improved target detection algorithm in complex railway environments using the RS dataset and the PASCAL VOC 2012 dataset [28]. The railway intrusion foreign object dataset (RS) is obtained from the internet and real images of railway surveillance. The three most frequently occurring targets in foreign object intrusion events, namely pedestrians, cars, and bicycles, are selected as the typical foreign objects for this paper. In total, 6000 images were used from the RS dataset, including 4000 images in the training set, 1000 images in the validation set, and 1000 images in the test set. The images in the training set and validation set are from the Internet, and the images in the test set are from real railway surveillance scenarios; the training set, validation set, and test set are relatively balanced in each category. The PASCAL VOC

2012 dataset contains four categories and 20 subcategories, with a total of 17,125 images, including 13,700 training images, 1713 verification images, and 1712 test images. Two samples of the datasets are shown in Figure 6.

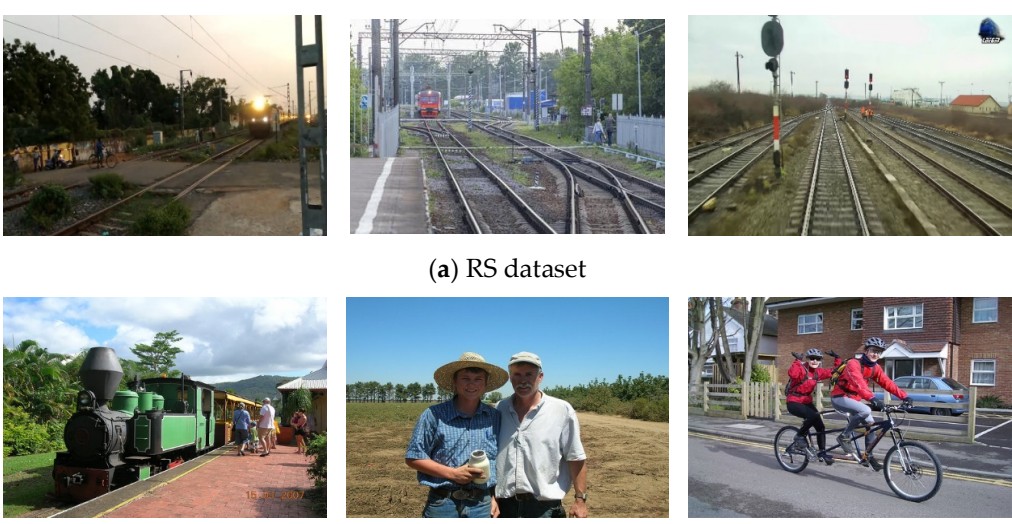

(**a**) RS dataset

(**b**) PASCAL VOC 2012 dataset

**Figure 6.** Dataset examples.

### 4.3. Evaluation Indicators

The evaluation indicators include precision (P), recall (R), mean average precision (mAP), parameters, giga-floating-point operations per second (GFLOPS), and frames per second (FPS). These are described by the following equations:

$$\text{Precision} = \frac{\text{TP}}{\text{TP} + \text{FP}} \tag{6}$$

$$\text{Recall} = \frac{\text{TP}}{\text{TP} + \text{FN}} \tag{7}$$

$$\text{mAP} = \frac{\sum_{i=1}^{N} \text{AP}_i}{N} \quad \text{AP} = \int_0^1 P(R) dR \tag{8}$$

$$\text{FPS} = \frac{\text{Total time}}{\text{NumFigure}} \tag{9}$$

TP means that the model predicts positive and accurate results; FP means positive and incorrect results; FN means negative and incorrect results. AP is the area under the PR curve. mAP measures the mean of all AP categories. mAP@0.5 represents the average accuracy when the IoU is set to 0.5, while mAP@0.5 : 0.95 represents the average accuracy when the IoU is from 0.5 to 0.95 and the step size is 0.05. FPS indicates the number of images that the network model can detect per second, namely, the network detection speed.

### 4.4. SSA Hybrid Attention Module Experimental Analysis

The hybrid attention module consists of the spatial attention submodule and SENet channel attention submodule in series; thus, the global information can play a complete role, better fit the relevant feature information, and improve the recognition effect of the model on small targets. We investigated the effects of different compositions of the channel attention and spatial attention modules and different serial orders in the SSA hybrid attention submodule on the RS dataset.

As shown in Table 1, the best results were obtained for spatial attention and channel attention SENet sequential serial performance, with mAP@0.5 and mAP@0.5:.95 increased by 1.1% and 1.5%, respectively. The reasoning speed and FPS of each image were also

increased, proving the effectiveness of the proposed hybrid attention module. Therefore, the hybrid attention composed of spatial attention and channel attention SENet is named SSA hybrid attention.

**Table 1.** Experimental results of different compositions and serial order in the attention submodule.

| Method | mAP@0.5 (%) | mAP@0.5:.95 (%) | Inference (ms) | FPS (bs = 1) |
|---|---|---|---|---|
| YOLOv5s (r6.1) | 0.84 | 0.583 | 10.3 | 86 |
| +(ECA-Spatial) | 0.839 (−0.1) | 0.59 (+0.7) | 11.5 | 66 |
| +(Spatial-ECA) | 0.84 (+0.0) | 0.585 (+0.2) | 11.3 | 75 |
| +(CA-Spatial) | 0.846 (+0.6) | 0.576 (−0.7) | 11.0 | 84 |
| +(Spatial-CA) | 0.844 (+0.4) | 0.589 (+0.6) | 10.9 | 74 |
| +(SENet-Spatial) | 0.838 (−0.2) | 0.585 (+0.2) | 10.0 | 78 |
| +(Spatial-SENet) | 0.851 (+1.1) | 0.598 (+1.5) | 9.8 | 90 |

We validated the effectiveness and generalization of the SSA hybrid attention module on the RS dataset and the PASCAL VOC 2012 dataset, and compared the effectiveness of SSA hybrid attention with commonly used attention modules. The experimental results are shown in Tables 2 and 3.

**Table 2.** Results of experiments on the RS dataset with various attention modules.

| Method | mAP@0.5 (%) | mAP@0.5:.95 (%) | Inference (ms) | FPS (bs = 1) |
|---|---|---|---|---|
| YOLOv5s (r6.1) | 0.84 | 0.583 | 10.3 | 86 |
| +SENet | 0.846 (+0.6) | 0.588 (+0.5) | 12.5 | 78 |
| +CA | 0.842 (+0.2) | 0.592 (+0.9) | 12.7 | 91 |
| +CBAM | 0.84 (+0.0) | 0.591 (+0.8) | 12.0 | 84 |
| +ECA | 0.848 (+0.8) | 0.58 (−0.3) | 12.1 | 74 |
| +SSA | 0.851 (+1.1) | 0.598 (+1.5) | 9.8 | 90 |

**Table 3.** Results of experiments on the PASCAL VOC 2012 dataset with various attention modules.

| Method | mAP@0.5 (%) | mAP@0.5:.95 (%) | Inference (ms) | FPS (bs = 1) |
|---|---|---|---|---|
| YOLOv5s (r6.1) | 0.674 | 0.462 | 9.7 | 79 |
| +SENet | 0.672 (−0.2) | 0.463 (+0.1) | 8.7 | 94 |
| +CA | 0.676 (+0.2) | 0.464 (+0.2) | 8.1 | 101 |
| +CBAM | 0.677 (+0.3) | 0.464 (+0.2) | 8.5 | 102 |
| +ECA | 0.677 (+0.3) | 0.466 (+0.4) | 9.5 | 90 |
| +SSA | 0.681 (+0.7) | 0.468 (+0.6) | 9.7 | 77 |

In comparative experiments on the RS dataset and PASCAL VOC 2012 dataset, the mAP@0.5 increased by 1.1% and 0.7%, and the mAP@0.5:.95 increased by 1.5% and 0.6%, respectively, while maintaining good detection speed. Thus, the effectiveness and wide applicability of the proposed SSA hybrid attention module were demonstrated.

By using a Grad-CAM (gradient-weighted class activation map) [29] to visually visualize the training weights of the proposed SDRC-YOLO network model, the effect of the SSA hybrid attention module is shown more intuitively. Each position in the output grid indicates the level of importance of that category. The heat maps of different attention modules were drawn by Grad-CAM to show the similarity between each position in the image and that category. Figure 7 compares the thermal map effect between the SSA hybrid attention mechanism proposed in this paper and the mainstream attention mechanism. It can be clearly seen that the thermal map of the focus area of the SSA hybrid attention mechanism on human targets is more concentrated; there is less focus on irrelevant features and there are not too many non-relevant feature areas, thus proving that SSA hybrid attention extracts positive and effective features.

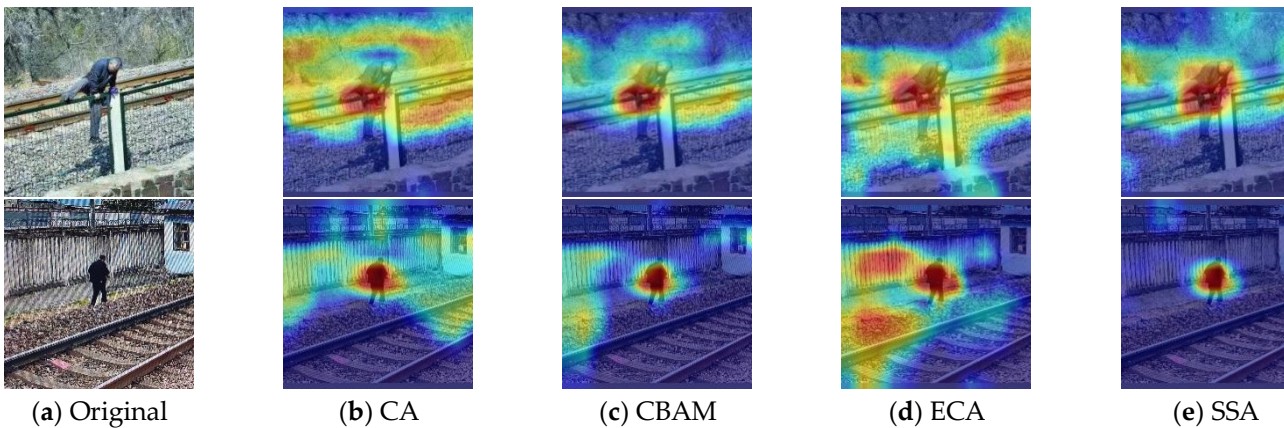

| (**a**) Original | (**b**) CA | (**c**) CBAM | (**d**) ECA | (**e**) SSA |

**Figure 7.** Heat map comparison of various attention mechanisms.

*4.5. Experimental Analysis of DW-Decoupled Head and Super Large Convolutional Kernel*

The DW-Decoupled Head is proposed to accelerate the convergence speed of the network, and a large convolutional kernel model RepLKNet is introduced to establish a larger sense field by using large convolutional kernels for feature extraction. Comparison experiments were set up on the RS dataset and the PASCAL VOC 2012 dataset confirmed the efficacy of the improved methods.

Tables 4 and 5 display the experimental results: the DW-Decoupled Head proposed enables the algorithm to improve the accuracy mAP@0.5 by 1.1 and 1.2 percentage points, and mAP@0.5:.95 by 0.3 and 2.1 percentage points, respectively, on the two data sets with the addition of a small number of parameters; in addition, by using a large convolutional kernel for feature extraction, the algorithm reduces the number of parameters. In addition, by using large convolutional kernels for feature extraction, the accuracy of the algorithm is enhanced by 1.2 and 1.8 percentage points for mAP@0.5 and by 0.3 and 3.1 percentage points for mAP@0.5:.95, respectively, and the detection speed remains the same as that of the original network, which indicates that the improvement improves the training convergence process and effect to a certain extent and proves the effectiveness and wide applicability of the improved methods.

**Table 4.** Results of two upgraded approaches' experiments on the RS dataset.

| Method | mAP@0.5 (%) | mAP@0.5:.95 (%) | Parameters (M) | GFLOPs | FPS (bs = 1) |
|---|---|---|---|---|---|
| YOLOv5s (r6.1) | 0.84 | 0.583 | 7.2 | 16.5 | 86 |
| +DW-Decoupled Head | 0.851 (+1.1) | 0.586 (+0.3) | 7.3 | 16.6 | 75 |
| +RepLKNet | 0.852 (+1.2) | 0.586 (+0.3) | 7.0 | 16.3 | 82 |

**Table 5.** Results of two upgraded approaches' experiments on the PASCAL VOC 2012 dataset.

| Method | mAP@0.5 (%) | mAP@0.5:.95 (%) | Parameters (M) | GFLOPs | FPS (bs = 1) |
|---|---|---|---|---|---|
| YOLOv5s (r6.1) | 0.674 | 0.462 | 7.0 | 16.1 | 79 |
| +DW-Decoupled Head | 0.686 (+1.2) | 0.483 (+2.1) | 7.3 | 16.8 | 72 |
| +RepLKNet | 0.692 (+1.8) | 0.493 (+3.1) | 6.7 | 15.6 | 85 |

*4.6. Experimental Analysis of CARAFE Lightweight Upsampling Operator*

The upsampling method of the algorithm is changed to the lightweight upsampling operator CARAFE. Figure 8 shows the comparison between the CARAFE upsampling feature graph and the nearest neighbor interpolation feature graph. It can be seen that CARAFE obtains smoother upsampling results and more reduced information. Experimental results are shown in Tables 6 and 7. The CARAFE upsampling structure can restore certain semantic information and location information, further improving the mAP@0.5 of results on the RS and PASCAL VOC 2012 datasets by 0.8% and 0.6%, respectively.

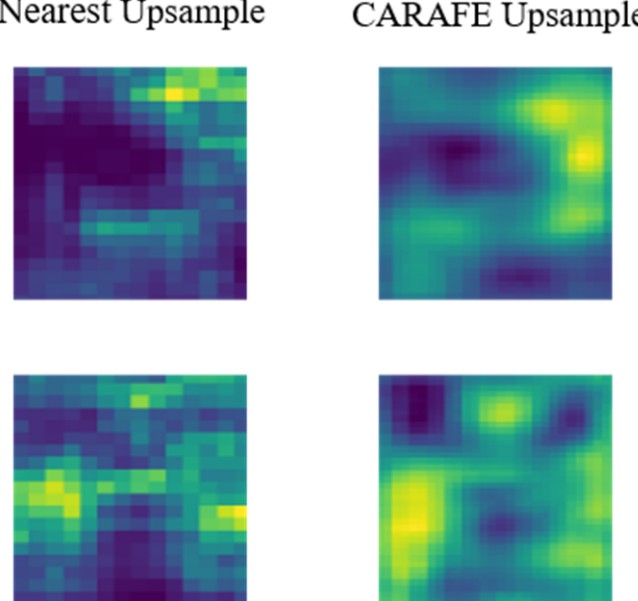

**Figure 8.** Comparison between the CARAFE upsample and the nearest upsample.

**Table 6.** Experimental results of CARAFE on the RS dataset.

| Method | mAP@0.5 (%) | mAP@0.5:.95 (%) | Parameters (M) | GFLOPs | FPS (bs = 1) |
|---|---|---|---|---|---|
| YOLOv5s (r6.1) | 0.84 | 0.583 | 7.2 | 16.5 | 86 |
| +CARAFE | 0.848 (+0.8) | 0.592 (+0.9) | 7.2 | 16.3 | 74 |

**Table 7.** Experimental results of CARAFE on the PASCAL VOC 2012 dataset.

| Method | mAP@0.5 (%) | mAP@0.5:.95 (%) | Parameters (M) | GFLOPs | FPS (bs = 1) |
|---|---|---|---|---|---|
| YOLOv5s (r6.1) | 0.674 | 0.462 | 7.0 | 16.1 | 79 |
| +CARAFE | 0.680 (+0.6) | 0.468 (+0.6) | 7.2 | 16.4 | 70 |

*4.7. Ablation Experiment*

SDRC-YOLO improves the attention mechanism, output feature approach, and up-sampling approach of the YOLOv5s model; ablation experiments were conducted on the RS and PASCAL VOC 2012 datasets in order to evaluate the different module changes and different module combinations in terms of algorithm performance optimization. The experimental ablation data are shown in Tables 8 and 9.

**Table 8.** Ablation experiments of SDRC-YOLO on the RS dataset.

| Method | mAP@0.5 (%) | mAP@0.5:.95 (%) | Parameters (M) | GFLOPs | FPS (bs = 1) |
|---|---|---|---|---|---|
| YOLOv5s (r6.1) | 0.84 | 0.583 | 7.2 | 16.5 | 86 |
| +RepLKNet (RLK) | 0.854 (+1.4) | 0.596 (+1.6) | 7.0 | 16.3 | 82 |
| +DW-DecoupledHead (DWHead) | 0.851 (+1.1) | 0.59 (+0.7) | 7.2 | 16.6 | 75 |
| +SSA | 0.851 (+1.1) | 0.598 (+1.5) | 7.0 | 15.8 | 90 |
| +CARAFE | 0.848 (+0.8) | 0.592 (+0.9) | 7.1 | 16.3 | 74 |
| +RLK + SSA + CARAFE | 0.866 (+2.6) | 0.602 (+1.9) | 7.1 | 16.5 | 81 |
| +RLK + SSA + DWHead | 0.861 (+2.1) | 0.601 (+1.8) | 7.3 | 16.3 | 75 |
| +RLK + DWHead + CARAFE | 0.864 (+2.4) | 0.598 (+1.5) | 7.3 | 16.5 | 83 |
| +SSA + DWHead + CARAFE | 0.858 (+1.8) | 0.602 (+1.9) | 7.3 | 16.5 | 79 |
| +RLK + SSA + DWHead + CARAFE | 0.868 (+2.8) | 0.606 (+2.3) | 7.3 | 16.5 | 80 |

**Table 9.** Ablation experiments of SDRC-YOLO on the PASCAL VOC 2012 dataset.

| Method | mAP@0.5 (%) | mAP@0.5:.95 (%) | Parameters (M) | GFLOPs | FPS (bs = 1) |
|---|---|---|---|---|---|
| YOLOv5s (r6.1) | 0.674 | 0.462 | 7.0 | 16.1 | 79 |
| +RepLKNet (RLK) | 0.692 (+1.8) | 0.493 (+3.1) | 6.7 | 15.6 | 85 |
| +DW-DecoupledHead (DWHead) | 0.684 (+1.0) | 0.474 (+1.2) | 7.3 | 16.8 | 80 |
| +SSA | 0.68 (+0.6) | 0.468 (+0.6) | 7.0 | 15.9 | 77 |
| +CARAFE | 0.68 (+0.6) | 0.468 (+0.6) | 7.2 | 16.4 | 70 |
| +RLK + SSA + CARAFE | 0.694 (+2.0) | 0.495 (+3.3) | 7.0 | 16.1 | 74 |
| +RLK + SSA + DWHead | 0.697 (+2.3) | 0.498 (+3.6) | 7.2 | 16.4 | 76 |
| +RLK + DWHead + CARAFE | 0.693 (+1.9) | 0.494 (+3.2) | 7.3 | 16.4 | 75 |
| +SSA + DWHead + CARAFE | 0.691 (+1.7) | 0.477 (+1.5) | 7.3 | 16.4 | 80 |
| +RLK + SSA + DWHead + CARAFE | 0.698 (+2.4) | 0.493 (+3.1) | 7.3 | 16.4 | 75 |

Whether in the RS dataset or the PASCAL VOC 2012 dataset, the four enhanced approaches in this study, to varying degrees, increased detection accuracy in comparison to the original YOLOv5s algorithm. Through the ablation experiment, the effectiveness of the combination of different improvement methods proposed in this paper was again proven. Meanwhile, compared with the original YOLOv5s, the final algorithm proposed in this paper improved mAP@0.5 by 2.8% and 2.4% on the RS dataset and PASCAL VOC 2012 datasets, respectively, while mAP@0.5:.95 was improved by 2.3% and 3.1%, respectively. The FPS values were up to 80 and 75, respectively, both of which can maintain real-time performance of the algorithm with higher detection accuracy. The algorithm proposed in this paper has good generalization and can better address the problem of target detection in different scene conditions under complex railway backgrounds.

*4.8. Comparison Experiment*

To further demonstrate the effectiveness and superiority of this proposal, we experimentally compared the final proposed algorithm model with the original YOLOv5s, YOLOv5m, SSD, Fast R-CNN, Faster R-CNN, YOLOv3, YOLOv4-tiny [30], YOLOv4 [30], YOLOX-tiny [31], YOLOX-S [31], YOLOv6-tiny [32], and YOLOv7-tiny [33] algorithms on the RS dataset. The results are shown in Table 10.

**Table 10.** Comparison experiments of different target detection algorithms on the RS dataset.

| Method | mAP@0.5 (%) | mAP@0.5:.95 (%) | Parameters (M) | GFLOPs | FPS (bs = 1) |
|---|---|---|---|---|---|
| YOLOv5s (r6.1) | 0.84 | 0.583 | 7.2 | 10.3 | 86 |
| YOLOv5m (r6.1) | 0.862 | 0.635 | 20.9 | 18.8 | 69 |
| SSD | 0.654 | 0.404 | 100.2 | 22.1 | 54 |
| Fast R-CNN | 0.685 | 0.426 | 56.3 | 18.1 | 34 |
| Faster R-CNN | 0.703 | 0.448 | 136.7 | 45.6 | 22 |
| YOLOv3-SPP | 0.823 | 0.544 | 9.56 | 11.3 | 80 |
| YOLOv4-tiny | 0.734 | 0.512 | 5.9 | 8.9 | 110 |
| YOLOv4 | 0.815 | 0.543 | 244.8 | 54.5 | 34 |
| YOLOX-tiny | 0.825 | 0.562 | 5.1 | 15.8 | 42 |
| YOLOX-S | 0.846 | 0.595 | 9.0 | 22.5 | 24 |
| YOLOv6-tiny | 0.862 | 0.604 | 15.0 | 21.5 | 80 |
| YOLOv7-tiny | 0.831 | 0.564 | 6.1 | 7.3 | 118 |
| SDRC-YOLO | 0.868 | 0.606 | 7.3 | 13.8 | 80 |

SDRC-YOLO has the highest detection accuracy when compared to other widely used algorithm models, according to a comparison of experimental results. Compared with the YOLOv5s algorithm and YOLOv3 algorithm, which have similar detection speed, the detection accuracy of the proposed algorithm in this paper is significantly superior: 2.8% higher than that of YOLOv5s and 4.5% higher than that of YOLOv3. Although YOLOv4-tiny and YOLOv7-tiny have higher detection speeds—110 FPS and 118 FPS—their detection accuracy is relatively low at 73.4% and 83.1%, not meeting the requirements of

the complex railway background. Compared with the YOLOv5s, YOLOv4-tiny, YOLOX-S, and YOLOX-tiny algorithms with similar parameters, the detection accuracy of the proposed algorithm is 2.8%, 13.4%, 2.2%, and 4.3% higher, respectively. The SDRC-YOLO algorithm suggested in this paper has the best detection accuracy, maintains real-time performance, and has exceptional overall performance, demonstrating its superiority over comparable algorithms.

A comparison of the detection effect before and after the improvements is presented to allow for a more intuitive assessment, as shown in Figure 9. SDRC-YOLO addresses the problems of false detection and missing detection in the detection of occluded targets and small targets with a complex railway background, according to comparative experimental results.

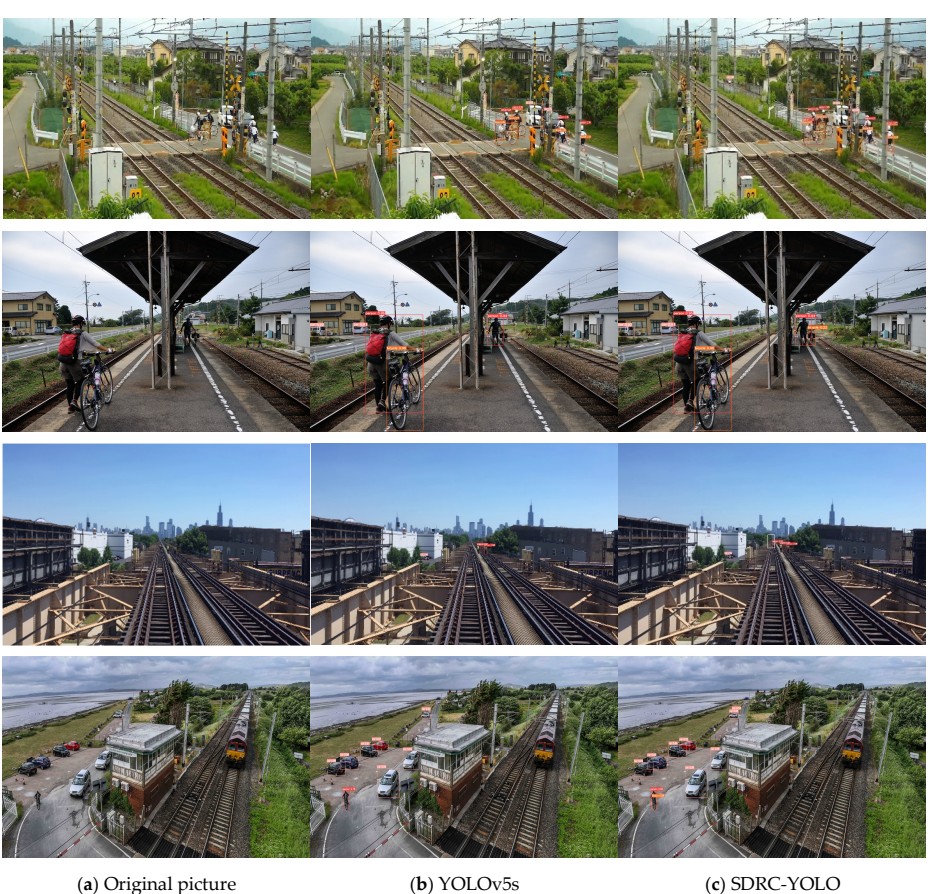

  (**a**) Original picture     (**b**) YOLOv5s     (**c**) SDRC-YOLO

**Figure 9.** Detection effect comparison of SDRC-YOLO and the original YOLOv5s.

## 5. Conclusions

To effectively reduce the problems of low accuracy and poor timeliness of foreign object intrusion detection methods in complex railway environments, this paper proposes SSA hybrid attention, DW-Decoupled Head, and foreign object intrusion detection in complex railway backgrounds by using an oversized convolution kernel and a lightweight upsampling operator. The experimental results demonstrate that the SDRC-YOLO algorithm improves mAP@0.5 by 2.8% and 2.4% on the homemade dataset RS and the public dataset PASCAL VOC 2012, respectively, with detection speeds of 80 FPS and 75 FPS. In comparison to other popular target detection algorithm models, we achieve higher detection accuracy while maintaining the algorithm speed and volume advantages. The improved algorithm is therefore better suited for foreign object intrusion detection in complex railway environments. In further study, achieving high detection accuracy with a lightweight network will be the focus of our research.

**Author Contributions:** Funding acquisition, L.S.; methodology, C.M.; supervision, L.S., Y.G., Y.T. and L.W.; writing—original draft, Z.W.; writing—review and editing, C.M. All authors have read and agreed to the published version of the manuscript.

**Funding:** This research was funded by National Key R&D Program of China (grant no. 2018YFB1701401, 2020YFB1712401-1), National Natural Science Foundation of China (grant no. 62006210, 62001284), Key Project of Public Benefit in Henan Province of China (grant no. 201300210500), and the Research Foundation for Advanced Talents of Zhengzhou University (grant no. 32340306).

**Data Availability Statement:** The data used to support the findings of this study are available from the corresponding author, wangzhaonan@gs.zzu.edu.cn (Z.W.), upon request.

**Conflicts of Interest:** The authors declare no conflict of interest.

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
