# Peer review of "SDRC-YOLO: A Novel Foreign Object Intrusion Detection Algorithm in Railway Scenarios"

_electronics, doi:10.3390/electronics12051256_

Round 1

Reviewer 1 Report

Respective authors have developed a novel foreign object intrusion detection algorithm in railway scenarios, however, according to the presented results they have focused on human detection as a foreign object. 
Therefore, please also show the algorithm's ability for other probable objects in railway scenarios like cars and tracks. 
Please also apply CNN as well as HOG to the data in order to have a better comparison. 
regards, 

Author Response

Thank you for pointing out problems in the manuscript.The manuscript and response to reviewers' comments are submitted as an attachment.

Reviewer 2 Report

In this paper, the authors proposed an efficient railway foreign object intrusion detection approach based on the SDRC-YOLO.

Overall:

-          The study and the topic are very interesting.

-          The work is well written and organized.

-          The experiments are well done and several metrics have been considered.

However:

-          Authors should define all acronyms.

-          Paper’s organization should be added in the last part of the manuscript.  

-          What assumptions authors made during the simulation phase of this research work? Provide a critique on this aspect.

-          Discuss some of the limitations that authors faced during this research work. Also authors need to highlight the research gap for existing works need to be explained.

-          Does this kind of study have never attempted before?  Justify this statement and give an appropriate explanation to do so in this paper.

Author Response

(The authors gave the same response as above.)

Reviewer 3 Report

The authors proposed an intrusion detection algorithm of foreign object in railway scenarios. In order for the submitted paper to be published in this journal, the authors must correct the following problems.

(1) The authors should explain the meaning of SDRC-YOLO in detail.

(2) In the experimental results of this paper, it is difficult to find a meaningful difference in performance comparison between the proposed method and the existing method, and a detailed explanation is needed.

Author Response

(The authors gave the same response as above.)

Reviewer 4 Report

• The introduction is not clear and very less literature is used. Follow this instruction: The introduction should briefly place the study in a broad context and highlight why it is important. It should define the purpose of the work and its significance, including specific hypotheses being tested. The current state of the research field should be reviewed carefully, and key publications cited. Please highlight controversial and diverging hypotheses when necessary. Finally, briefly mention the main aim of the work and highlight the main conclusions. Keep the introduction comprehensible to scientists working outside the topic of the paper.

• In the introduction, what key theoretical perspectives and empirical findings in the main literature have already informed the problem formulation? What major, unaddressed puzzle, controversy, or paradox does this research address? 

• It would be interesting if the authors report the trade-off compared to other methods especially the computational complexity of the models. Some techniques require more memory space and take longer time, please elaborate on that. 

• Authors should further clarify and elaborate novelty in their contribution.

• What are the limitations of the present work?

• Below papers has some interesting implications that you could discuss in your introduction and how it relates to your work.

1) Basak, H., Kundu, R., Singh, P.K. et al. A union of deep learning and swarm-based optimization for 3D human action recognition. Sci Rep 12, 5494 (2022). https://doi.org/10.1038/s41598-022-09293-8

2) Hussain, R.; Karbhari, Y.; Ijaz, M.F.; Woźniak, M.; Singh, P.K.; Sarkar, R. Revise-Net: Exploiting Reverse Attention Mechanism for Salient Object Detection. Remote Sens. 2021, 13, 4941. https://doi.org/10.3390/rs13234941

Author Response

(The authors gave the same response as above.)

Round 2

Reviewer 2 Report

The manuscript can be accepted in the current form.

Reviewer 4 Report

.